# Reservoirs and transmission routes of leprosy; A systematic review

**Thomas Ploemacher**[1], **William R. Faber**[2], **Henk Menke**[1], **Victor Rutten**[3,4],
**Toine Pieters**[1] *

**1** Faculty of Science, Freudenthal Institute & Utrecht Institute for Pharmaceutical Sciences (UIPS), Utrecht University, Utrecht, the Netherlands, **2** Faculty of Medicine, Department of Dermatology, University of Amsterdam, Amsterdam, the Netherlands, **3** Faculty of Veterinary Medicine, Utrecht University, the Netherlands, **4** Dept of Veterinary Tropical Diseases, Faculty of Veterinary Science, University of Pretoria, Republic of South Africa

* t.pieters@uu.nl

**Data Availability Statement:** All relevant data are within the manuscript and its Supporting Information files.

**Funding:** The author(s) received no specific funding for this work.

## Abstract

Leprosy is a chronic infectious disease caused by *Mycobacterium leprae (M. leprae)* and the more recently discovered *Mycobacterium lepromatosis (M. lepromatosis)*. The two leprosy bacilli cause similar pathologic conditions. They primarily target the skin and the peripheral nervous system. Currently it is considered a Neglected Tropical Disease, being endemic in specific locations within countries of the Americas, Asia, and Africa, while in Europe it is only rarely reported. The reason for a spatial inequality in the prevalence of leprosy in so-called endemic pockets within a country is still largely unexplained. A systematic review was conducted targeting leprosy transmission research data, using PubMed and Scopus as sources. Publications between January 1, 1945 and July 1, 2019 were included. The transmission pathways of *M. leprae* are not fully understood. Solid evidence exists of an increased risk for individuals living in close contact with leprosy patients, most likely through infectious aerosols, created by coughing and sneezing, but possibly also through direct contact. However, this systematic review underscores that human-to-human transmission is not the only way leprosy can be acquired. The transmission of this disease is probably much more complicated than was thought before. In the Americas, the nine-banded armadillo (*Dasypus novemcinctus*) has been established as another natural host and reservoir of *M. leprae*. Anthroponotic and zoonotic transmission have both been proposed as modes of contracting the disease, based on data showing identical *M. leprae* strains shared between humans and armadillos. More recently, in red squirrels (*Sciurus vulgaris*) with leprosy-like lesions in the British Isles *M. leprae* and *M. lepromatosis* DNA was detected. This finding was unexpected, because leprosy is considered a disease of humans (with the exception of the armadillo), and because it was thought that leprosy (and *M. leprae*) had disappeared from the United Kingdom. Furthermore, animals can be affected by other leprosy-like diseases, caused by pathogens phylogenetically closely related to *M. leprae*. These mycobacteria have been proposed to be grouped as a *M. leprae*-complex. We argue that insights from the transmission and reservoirs of members of the *M. leprae*-complex might be relevant for leprosy research. A better understanding of possible animal or environmental reservoirs is needed, because transmission from such reservoirs may partly explain the steady

**Competing interests:** The authors have declared that no competing interests exist.

global incidence of leprosy despite effective and widespread multidrug therapy. A reduction in transmission cannot be expected to be accomplished by actions or interventions from the human healthcare domain alone, as the mechanisms involved are complex. Therefore, to increase our understanding of the intricate picture of leprosy transmission, we propose a *One Health* transdisciplinary research approach.

## Author summary

Leprosy is a chronic infectious disease caused by *Mycobacterium leprae (M. leprae)* and the more recently discovered *Mycobacterium lepromatosis (M. lepromatosis)*. The two leprosy bacilli cause similar stigmatizing pathologic conditions. *M. leprae* primarily targets the skin and the peripheral nervous system. Currently it is considered a Neglected Tropical Disease. The transmission pathways of *M. leprae* are not fully understood. Solid evidence exists of an increased risk for individuals living in close contact with leprosy patients, most likely through infectious aerosols, created by coughing and sneezing, but possibly also through direct contact. However, this systematic review underscores that human-to-human transmission is not the only way leprosy can be acquired. Anthroponotic and zoonotic transmission have both been proposed as modes of contracting the disease, based on data showing identical *M. leprae* strains shared between humans and armadillos. A better understanding of possible animal or environmental reservoirs is needed, because transmission from such reservoirs may partly explain the steady global incidence of leprosy despite effective and widespread multidrug therapy. Reducing transmission cannot be expected from the human healthcare domain alone, as the mechanisms involved are complex. Therefore, we propose a *One Health* transdisciplinary research approach.

## Introduction

Leprosy, also called Hansen's disease, results from infection with *Mycobacterium leprae (M. leprae)* or *Mycobacterium lepromatosis (M. lepromatosis)*. It is a chronic infectious disease which primarily affects the skin and peripheral nerves. It ranges from a localized to a systemic infection. Damage of peripheral nerves can lead to serious impairment and disability.[1] Transmission pathways of *M. leprae* are not fully understood. Solid evidence exists of an increased risk for individuals living in close contact with leprosy patients, most likely through infectious aerosols, created by coughing and sneezing, but possibly also through skin to skin contact.[2,3] Multidrug treatment (MDT) developed in the 1980s is an effective chemotherapy and has played a critical role in the worldwide reduction of the leprosy burden, by reducing human-to-human transmission.[4] However, despite widespread application of this therapy, the World Health Organization recorded 208,619 new leprosy cases globally in 2018, only slightly less than the 219,075 cases in 2011.[5] Even though leprosy has been eliminated in most countries, in certain areas endemic leprosy persists.

Hansen (1874) was the first to report rod-shaped bodies resembling bacteria in cells from leprosy patients using a light microscope.[6] Before that, leprosy was thought to be of environmental or hereditary nature.[7] After the discovery by Hansen, attempts were made to grow the pathogen on an array of artificial media and in numerous animals with the purpose of studying its characteristics, and potential treatments. With an improved tissue staining

technique called Ziehl-Neelsen (ZN) it became possible to detect and visualize acid fast bacilli (AFB); and the Wade-Fite modification for the demonstration of *M.leprae*. In cases of leprosy, unspecified non-cultivable AFB were found.

*M. leprae* is an obligate intracellular pathogen that has never been cultured in vitro but can be cultivated in vivo in experimental animals. Shepard demonstrated in the early 1960s that the mouse footpad (MFP) could be infected with *M. leprae*.[8] Characteristic of this model are the local infection and the slow replication rate.[8,9] Armadillos had been used successfully in medical research, and their cool body temperature of 32–35˚C attracted the attention of leprosy researchers. In 1971 Kirchheimer and Storrs, working in Louisiana (USA), reported the first successful experimental infection of the nine-banded armadillo with *M. leprae*.[10,11]

Until 2008, leprosy was thought to be caused exclusively by *M. leprae*. Analysis of leprosy patients in Mexico by Han et al. showed that a second species, *M. lepromatosis*, causes leprosy as well.[12]

*M. leprae* and *M. lepromatosis* diverged from a common ancestor about 13.9 million years ago (Mya). The two pathogens are similar in genome size ($\sim$ 3,27 MB), and were subject to extensive reductive evolution.[12,13] Protein-coding genes share 93% nucleotide sequence identity.[14] Using genome scale comparison, the worldwide distribution and evolution of *M. leprae* have been studied. Phylogenetically, at least four SNP types (1–4) and five branches (0–4) and 16 SNP subtypes (A to P) have been distinguished. Global distribution is thought to have coincided with human migration.[15,16]

In medieval times, leprosy was endemic all over Eurasia.[17] Currently, this so-called Neglected Tropical Disease is endemic in regions within countries of the Americas, Asia, and Africa. In Europe, it is now only rarely reported, and infection is acquired outside Europe. The reasons for a spatial inequality of leprosy in endemic pockets within a country are still largely unexplained.

Further research is required to clarify transmission mechanisms and natural reservoirs of leprosy pathogens.[18] In the Americas, the nine-banded armadillo (*Dasypus novemcinctus*) has been established as a natural host and reservoir of *M. leprae*, and identical *M. leprae* strains are shared between humans and armadillos.[19,20] A reservoir can be defined with respect to a target population as 'one or more epidemiologically connected populations or environments in which a pathogen can be permanently maintained and from which infection is transmitted to the target population'.[21] Hence, both anthroponotic and zoonotic transmission have been proposed as modes of transmission. However, Scollard argued recently that there is a general bias towards human-to-human transmission of leprosy.[22]

In 2016, both *M. leprae* and *M. lepromatosis* DNA was found in red squirrels (*Sciurus vulgaris)* with leprosy-like lesions in the British Isles.[23] This was highly unexpected, primarily because leprosy was considered to be a disease of humans (with the exception of the armadillo), and secondly, because *M. leprae* was thought to have disappeared from the United Kingdom. The *M. leprae* strain isolated from red squirrels is essentially the same as the one that circulated in humans in medieval England and Denmark, and is closely related to the strains carried by armadillos in the southern United States.[23] Transmission routes from animal reservoirs are possibly interrelated with the environment in an elaborate ecological cycle, which has yet to be unraveled. This may partly explain the steady global incidence of leprosy, as effective and widespread multidrug therapy reduces only human-to-human transmission.

This systematic review aims to provide an overview of worldwide research regarding non-human reservoirs and transmission. Potential natural reservoirs and additional transmission routes of leprosy are discussed.

## Method

A systematic review was conducted targeting leprosy transmission research data using PubMed and Scopus as sources. The review protocol was not pre-registered at PROSPERO. The search strategy was composed of three topics: wildlife research, environmental research, and studies on zoonosis and vectors of leprosy. Search strategies have been listed in the PRISMA flowchart (Fig 1). PRISMA stands for Preferred Reporting Items for Systematic Reviews and Meta-Analyses. It is an evidence-based minimum set of items for reporting in systematic reviews and meta-analyses.[24]

Wildlife research looked to find studies on known reservoirs, the armadillo and the red squirrel and prevalence, and on new reservoirs. Environmental research revolved around the possibility of survival of *M. leprae* or *M. lepromatosis* in extracellular conditions. Knowledge on the transmission of leprosy is limited, and any evidence for leprosy transmission by animals or insects was targeted through the search on zoonosis and vectors.

Publications between January 1, 1945 and July 1, 2019 were included. Articles that appeared with a title translated to English qualified for inclusion. Translations from Portuguese were obtained using an online translation service, and a Chinese article was translated with the help of a native colleague. This enabled an inclusion process as described in Fig 1. Three searches were performed, on leprosy in wildlife, environmental sources of leprosy, and leprosy zoonosis and vectors, using the following search strings.

Scopus, wildlife: TITLE-ABS-KEY (leprosy) OR TITLE-ABS-KEY (m.leprae) OR TITLE-ABS-KEY (m.lepromatosis) AND TITLE-ABS-KEY (wildlife) OR TITLE-ABS-KEY (animal).

PubMed, wildlife: (Leprosy OR M.leprae OR M.lepromatosis) AND (wildlife OR animal)

Scopus, environment: TITLE-ABS-KEY (m.leprae OR m.lepromatosis OR leprosy ) AND TITLE-ABS-KEY (environment OR ecology OR soil OR water ).

PubMed, environment: (leprosy OR m.leprae OR m.lepromatosis) AND (environment OR ecology OR soil OR water)

Scopus, vectors: TITLE-ABS-KEY (m.leprae OR m.leprae OR leprosy ) AND TITLE-ABS-KEY (zoonosis OR vector OR insect OR arthropod OR amoeba).

PubMed, vectors: (leprosy OR m.leprae OR m.lepromatosis) AND (zoonosis OR vector OR insect OR arthropod OR amoeba).

Articles from the search were included by TP (first author) based on title, then on abstract, and finally on a full-text assessment. Articles were included when any relevance to the primary subject of the search was assumed. Exclusion criteria were used to reduce the number of irrelevant studies. For the wildlife search subject, studies on experimental inoculation of animals, studies performed *ex vivo*, and drug animal studies were rejected. For the other search subjects, inclusion for relevance based on title and abstract was less strict to reduce loss of relevant abstracts. *In vitro* studies were accepted, for instance, because these studies could provide relevant evidence for the survival of *M. leprae* outside a host. Fig 1 shows the number of articles selected and rejected during the search process. 81 of the articles used in this article were selected in this way. To maximize finding eligible articles, the citations and references of included articles and related reviews were investigated ("snowballing"). This yielded an additional 64 articles.

Data extracted in the results were predominantly qualitative. Quantitative data were copied from the full texts. In some cases, when percentages were not available in the texts, they were calculated. Some prevalence data of armadillos were pooled. This was done to reduce the prevalence reporting in this systematic review, instead of mentioning prevalence in each region. No meta-analysis has been performed with quantitative data.

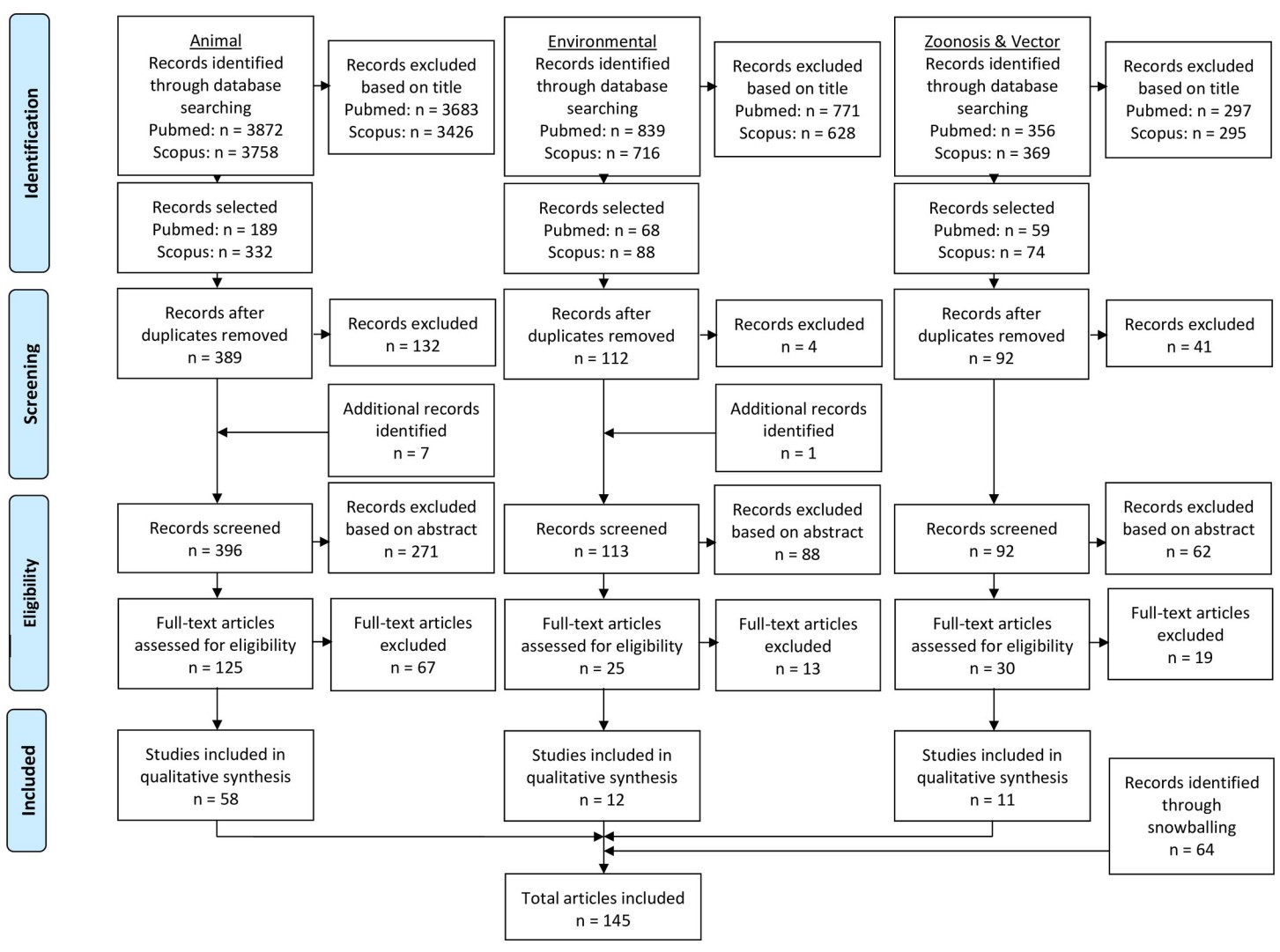

**Fig 1.**

Methods of studies and result interpretation were considered, but risk of bias was not systematically scored. A discussion of the limitations of some studies is incorporated where these could have influenced the findings or conclusions made. It was not possible to formally assess publication bias due to the variation of study methods. For the PRISMA checklist accompanying the PRISMA flow diagram of the search we refer to the online supplementary data: S1 PRISMA Checklist.

## Results

### Wildlife reservoirs

**The nine-banded armadillo in the southern United States (see Table 1).**   In 1975, Walsh et al. reported seven nine-banded armadillos with leprosy-like clinical signs from different locations in Louisiana.[25] In a large scale follow-up study 1033 armadillos from different locations in Louisiana as well as the neighboring states Texas, Florida and Mississippi were examined.[26] A total of 50 animals (4.8%) were found with naturally acquired leprosy-like disease.

**Table 1. Prevalence of *M. leprae* in wildlife 9-banded armadillos in the United States.**

| Authors (year of publication) | Applied methods | | | | Location(s) | Results |
|---|---|---|---|---|---|---|
| | Samples | AFB staining | PGL I | PCR | | |
| Walsh et al. (1977) [26] | Road kills and caught armadillos (not specified). Ear-tissue samples. | Yes | NA | NA | Louisiana | 49/691 (7,09%) |
| | | | | | Texas | 1/88 (1.14%) |
| | | | | | Florida | 0/76 |
| | | | | | Mississippi | 0/178 |
| Skinsnes (1976)[30] | Lymph nodes, spleen, and liver (n-89). Blood buffy coat and ear-clip (n = 144). | Yes | NA | NA | Louisiana | 0/133 |
| | | | | | Texas | 0/13 |
| | | | | | Florida | 0/87 |
| Smith et al. (1978) [41] | Ear tissue smears. Autopsy: Liver, spleen, lymph nodes, lepromas, omentum. | Yes | NA | NA | Louisiana | 5/20 (25%) |
| Smith et al. (1983) [36] | Ear tissue (n-451). Autopsy upon suspicion of leprosy. | Yes | NA | NA | Texas | 21/451 (4.66%) |
| Job et al. (1986)[37] | Ear biopsy of road killed armadillos (n = 494). | Yes | No | NA | Louisiana | 10/494 (2.02%) |
| Clark et al. (1987) [38] | Ear tissue of road killed (n = 213), captive (n = 12) and shot (n = 12) armadillos. | Yes | No | NA | Texas | 0/237 |
| Stallknecht et al. (1987)[39] | 67 sera and 74 ear sections from captured armadillos (n = 77). | Yes | Yes | NA | Louisiana | • PGL I 5/67 (7.5%)<br>• AFB 1/74 (1.3%) |
| Howerth et al. 1990 [40] | Ear samples from road killed (n≈700), shot or captured armadillos (total n = 853). | Yes | No | NA | Alabama (n = 144), Arkansas (n = 60), Florida (n = 93), Georgia (n = 246) and Mississippi (n = 310). | 0/853 |
| Truman et al. (1991) [43] | Blood samples. | Yes | Yes | NA | Louisiana | • PGLI 84/530 (15.8%)<br>• AFB 17/493 (3.4%) |
| | | | | | Texas | • PGLI 6/35 (17.1%)<br>• AFB 2/35 (5.7%) |
| Paige et al. (2002) [45] | Blood samples (1987–1989, 1997). Ear tissue for staining (1997). | Yes | Yes | No | Louisiana | • PGLI 79/414 (19%, 90%CI 18.4–19.7%).<br>• AFB 5/165 (3.0%). |
| Morgan and Loughry (2009)[46] | Blood samples of adult and young armadillos in 2007 and 2008. | No | Yes | No | Mississippi | • Adult: 32/210 (15.2%)<br>• Young: 0/75 |
| Loughry et al. (2009) [47] | Serum or blood. Ear or spleen tissue for PCR if PGL I positive. | No | Yes | Yes | Mississippi | 12/259 (4.6%) |
| | | | | | Georgia | 0/65 |
| | | | | | Alabama | 8/138 (5.8%) |
| Williams and Loughry (2012)[48] | Serum or blood. Ear tissue biopsy for PCR. (2005–2010) | No | Yes | Yes | Mississippi | 86/930 (9.2% 95%CI 5.5–13.0%) |
| Sharma et al. (2015) [19] | Serum or whole blood (n = 645). PCR of lymph tissue (if available) when positive for PGL I (n = 95). | No | Yes | Yes | Alabama | • PGLI 65/436 (14.9%)<br>• PCR 61/61 |
| | | | | | Georgia | • PGLI 27/148 (18.2%)<br>• PCR 22/22 |
| | | | | | Florida | • PGLI 7/23 (30.4%)<br>• PCR 7/7 |
| | | | | | Mississippi | • PGLI 7/38 (18.4%)<br>• PCR 5/5 |
| Perez-Heydrich (2016)[49] | Blood samples (2005–2010). | No. | Yes. | No. | Mississippi | PGLI 88/838 (10.5%) |

*AFB: Acid Fast Bacilli

**Mouse footpad method: inoculation into mouse footpad of *M. leprae* results in a local infection with a distinct growth curve characteristic of *M. leprae*.

The prevalence varied from 0% to 29.6%, depending on the location where armadillos were caught. On average, the prevalence in Louisiana was 7.1% (n = 691) and in Texas 1.1% (n = 88). The histopathological pattern and the presence of acid fast bacilli nine-banded armadillos resembled lepromatous leprosy, and suggested that it was caused by *M. leprae*.[27,28]

There was scientific controversy about the origin of leprosy in wild armadillos.[29] The possibility of laboratory spillover was considered, and the absence of mycobacterial infection in armadillos from the same region was reported.[30,31] Clinical and epidemiological investigations defined the disease in these armadillos, and confirmed its identity with human leprosy. [32,33] The concern of the possible wildlife contamination caused by experimentally infected armadillos addressed in 1986 by Truman et al. using a new technique. Phenolic glycolipid I (PGLI) is a *M. leprae* specific antigen, and by an enzyme-linked immunosorbent assay (ELISA) anti-PGL I antibodies can be detected in serum samples.[34] They tested stored blood samples from wild nine-banded armadillo which had been collected between 1960 and 1964, prior to the first reported experimental infection of armadillos in 1971. Positive samples confirmed that leprosy infection in armadillos predated the laboratory experiments.[35]

Gradually, the extent of natural infection of armadillos with *M. leprae* in the southern United States (Alabama, Arkansas, Florida, Georgia, Louisiana, Mississippi and Texas) became apparent with infection rates between 0% and 7,5%.[36–41] Truman et al. (1990) showed a discrepancy between histopathological findings and serological findings in an armadillo study population.[42] Histopathological presence of AFB in the ear was associated with late stage and disseminated disease, while early stages could be detected by anti-PGL I antibodies in serum. This was shown in 216 nine-banded armadillos captured in Louisiana: 12.3% of these animals were positive for anti-PGL I, while only 2.7% of the armadillos were positive on histopathological examination. Truman et al (1991) proceeded with the investigation of 565 nine-banded armadillos from three sites in Louisiana and one site in Texas. Of the 35 Texan armadillos, 17.1% had anti-PGL I antibodies, and 5.7% had AFB. Of the 530 Louisiana armadillos, 15.8% had anti-PGL I antibodies, and 3.4% had AFB.[43]

With the introduction of the Polymerase Chain Reaction (PCR) technique, new investigations were initiated. Job et al. compared PCR detection rates with other methods for detection of *M. leprae*.[44] Thirty armadillos were investigated. PCR of lymph nodes showed that 16 animals (53.3%) were infected with *M. leprae*, while a positive ELISA for anti-PGL1 and lepromatous granuloma were found in 2 (6.7%). Disease signs were found at autopsy in 13.3% of these armadillos. In 2002, Paige et al. studied the prevalence of and a possible relationship between *Trypanosoma cruzi* and *M. leprae* infection in 415 wild nine-banded armadillos from Louisiana.[45] Antibodies against pathogen specific antigens were detected using ELISA. The average prevalence of antibodies against *M. leprae* was 19% and against *T. cruzi* 3.9%. Morgan and Loughry (2009) studied nine-banded armadillos in western Mississippi and could not find clinical signs of leprosy in young or juvenile armadillos, suggesting no vertical transmission. [46] Loughry et al. (2009) studied the possible spread of *M. leprae* eastward from Mississippi, Alabama and Georgia into Florida.[47] Armadillo samples were examined for anti-PGL I antibodies and PCR for the detection of *M. leprae*. Infection prevalence ranged from 0–10%. In the eastern sampling sites that were considered *M. leprae* free, a few infected armadillos were found. They concluded that *M. leprae* was spreading eastward, although prevalence rates were still much lower than in the populations west of the Mississippi river. According to Williams and Loughry (2012) the prevalence of infected armadillos in the southern United States, measured by anti-PGLI antibodies ranged from 4.5% in 2005 to 14.9% in 2010, with an average of 9.3% over a six year period.[48] Perez-Heydrich et al. (2016) studied demographic and spatial risk factors for *M. leprae* exposure of nine-banded armadillos in Western Mississippi.[49] GPS locations of the armadillos were determined by tagging armadillos. Over a period of six years

(2005–2010), 74 of 466 animals (15.9%) were seropositive for anti-PGL I. No spatial risk factors were found, and the study indicated that *M. leprae* infection is spatially homogeneous in the armadillo population in Western Mississippi.

**Armadillos in other parts of the Americas (see Table 2).** The presence of *M. leprae* and leprosy in wild nine-banded armadillos was discovered in the southern United States, and most studies on its prevalence have been performed there. Armadillos are common throughout the southern United States nowadays, but their geographic range extends from northern Argentina up to Mexico. They migrated from Mexico into the United States.[20] Only the nine-banded armadillo is common in the southern United States, but other armadillo species are indigenous to South America. Whereas wildlife studies almost exclusively focused on the nine-banded armadillo, susceptibility for *M. leprae* infection by experimental inoculation was also observed for other armadillo species like the Venezuelan or Llanos long-nosed armadillo (*Dasypus sabanicola*) [50,51] and the seven-banded armadillo (*Dasypus hybridus*).[52,53] In 1984, Amezcua et al. reported the occurrence of a wild armadillo infected with *M. leprae* in Mexico.[54] They found acid-fast bacilli in ears, tongue and lymph nodes. Homogenates of infected nodules and lymph nodes did not grow on Löwenstein-Jensen and 7H10 media. Mouse footpad inoculation using skin and lymph nodule suspensions showed growth comparable to inoculation with *M. leprae*. Martinez and colleagues reported a captured nine-banded armadillo in Argentina with mycobacteriosis with nerve involvement.[55] Staining and biochemical analysis as well as mouse footpad inoculation strongly suggested *M. leprae*. Deps et al. (2007, 2008) studied the presence of anti-PGL-I antibodies in Brazilian armadillos from the state of Espirito Santo. In their 2007 study, 29.7% of animals (n = 37) were positive, and in their 2008 study, 10.6% of the animals (n = 47) were positive.[56,57] Frota et al. (2012) examined 27 nine-banded armadillos, and 2 six-banded armadillos in the state of Ceará, Northeast Brazil, a leprosy endemic area.[58] Samples from the ear, nose, liver and spleen were examined by a nested *M. leprae*-specific repetitive element (RLEP) PCR assay. Five of the nine-banded armadillos, and 1 six-banded armadillo were positive for *M. leprae* specific DNA.

In 2009, Deem et al. reported on three-banded (*Tolypeutes matacus*) and nine-banded armadillos in the Gran Chaco, Bolivia.[60] Three-banded (n = 2) and nine-banded (n = 8) armadillos were negative for anti-PGLI antibodies. Cardona-Castro et al. (2009) examined armadillos in the forest areas surrounding cities with leprosy patients in Colombia during a two year period.[61] Ear lobe biopsies from 22 nine-banded armadillos were PCR analyzed for the *M. leprae-specific* repetitive element (*RLEP)* region. PCR was positive in 40.9% (9 of 22) of the armadillos.

Pedrini et al. (2010) examined 44 armadillos, from the mid-Western area of the state of São Paulo, and the Pantanal Floodplain of the state of Mato Grosso do Sul in Brasil.[62] They examined 18 nine-banded armadillos (*D. novemcinctus*), 22 six-banded armadillos (*Euphractus sexcinctus*), two greater naked-tailed armadillos (*Cabassous tatouay*), and two Southern naked-tailed armadillos (*Cabassous unicinctus*). In addition, 10 road killed animals were also analyzed. None of the samples were positive for *M. leprae* by PCR.

**Squirrels (see Table 3).** Experiments showed that ground squirrels (*Ictidomys tridecemlineatus*) were susceptible to infection with *M. leprae*. Galletti et al. (1982) observed AFB counts in these squirrels that indicated multiplication under hibernation inducing conditions.[63] Recent studies have shown that red squirrels (*Sciurus vulgaris*) in England and Scotland (where the endangered red squirrel is competing with the non-native gray squirrel from North America) are infected with *M. lepromatosis* or *M. leprae*. Meredith et al. (2014) reported a novel presentation of a dermatitis in 6 red squirrels from various locations throughout Scotland. All animals showed bilateral areas of variable alopecia and cutaneous swelling of the snout area, lips, eyelids, pinnae and the distal parts of all limbs.[64] Histological examination

**Table 2. Prevalence of *M. leprae* in wildlife armadillos in other parts of the Americas.**

| Authors (year of publication) | Applied methods | | | | | | Location(s) | Armadillo species | Results |
|---|---|---|---|---|---|---|---|---|---|
| | Samples | AFB | MFP | PGL I | PCR | | | | |
| Amezcua et al. (1984) [54] | Case report (n = 1). Ear, tongue, nasal and lymph node smears. Lymph tissue for MFP. 9-banded armadillo. | Yes. | Yes. | NA | NA | | State of Mexico, Mexico. | Nine-banded | • MFP positive • Staining positive. |
| Martinez and colleagues (1984)[55] | Case report (n = 1). Lesion derived samples. | Yes | Yes | NA | NA | | North East Argentina. | Nine-banded | • MFP positive • Staining positive. |
| Deps et al. (2007)[56] | Blood samples. | No. | No. | Yes | No. | | State of Espírito Santo, Brazil. | Nine-banded | PGL I: 11/37 (29.7%) |
| Deps et al. (2008)[59] | Blood samples (collected 2001–2002). | No. | No. | Yes | No. | | State of Espírito Santo, Brazil. | Nine-banded | PGL I: 5/47 (10.6%) |
| Deem et al. (2009)[60] | Blood samples. | No | No | Yes | No | | The Gran Chaco, Bolivia. | Nine-banded | PGL I: 0/2 |
| | | | | | | | | Three-banded | PGL I: 0/8 |
| Cardona-Castro et al. (2009)[62] | Ear lobe biopsy. | No | No | No | Yes | | Barbosa municipality, Colombia. | Nine-banded | PCR: 9/22 (40.9%) |
| Pedrini et al. (2010) [61] | Variety of samples: organs/tissues , feces, nostril swab, blood. | Yes | No | No | Yes | | State of São Paulo, Brazil. | Nine-banded (n = 18) | • PCR: 0/44 • AFB: 0/44 |
| | | | | | | | | Six-banded (n = 22) | |
| | | | | | | | | Great naked-tailed (n = 2) | |
| | | | | | | | | Southern naked-tailed (n = 2) | |
| Frota et al. (2012)[58] | Ear, nose, liver and spleen. | No | No | No | Yes | | State of Ceará, Brazil. | Nine-banded | 5/27 (18.5%) |
| | | | | | | | | Six-banded | 1/2 (50%) |

*AFB: Acid Fast Bacilli

**Mouse footpad method: inoculation into mouse footpad of *M. leprae* results in a local infection with a distinct growth curve characteristic of *M. leprae*.

on three squirrels showed granulomatous dermatitis, and epithelioid macrophages forming sheets, in addition to large numbers of AFB. PCR of various dermatitis causing pathogens was performed. Sequencing of the hsp65 PCR amplicons from the squirrels revealed 99% sequence homology with *M. lepromatosis* (FJ924). The exact nature of the mycobacterium involved and the characterization required further research. In view of these findings, another British research group re-examined four stored red squirrels with epidermal hyperplasia originating from the Isle of Wight and Brownsea Island. The presence of *M. lepromatosis* was confirmed by PCR in all four samples.[65]

A follow-up study was conducted to map the magnitude of infection in the UK.[23] The study also first reported neural involvement in red squirrels infected with *M. leprae* (n = 8) and *M. lepromatosis* (n = 4). Samples from 13 squirrels with and 101 animals without leprosy-like symptoms were examined using differential PCR for *M. leprae* and *M. lepromatosis*. *M. lepromatosis* was found in red squirrels from Scotland (6/44), Ireland (2/39), and the Isle of Wight (1/1).[66] All squirrels from Brownsea Island (n = 25) were positive for *M. leprae*. 21 squirrels without clinical signs and all 13 animals with clinical signs were PCR positive. Anti-PGLI ELISA was positive in a significant number of the red squirrel samples (see Table 3). Gray squirrels were also tested by PCR (n = 3) and for anti-PGLI antibodies (n = 4), but all samples were negative. Phylogenetic comparison of British and Irish *M. lepromatosis* with two Mexican *M. lepromatosis* strains from humans showed that they diverged from a common

**Table 3. Prevalence of *M. leprae* and *M. lepromatosis* in wild squirrels.**

| Authors (year of publication) | Methods applied | | | | Location(s) | Species | Result(s) |
|---|---|---|---|---|---|---|---|
| | Samples | AFB staining | Anti-PGL-I | PCR | | | |
| **Meredith et al. (2014)[64]** | Not specified. PCR hsp65. | Yes | No | Yes | Scotland | Red squirrels | • AFB: 3/3 positive • PCR: 3/3 |
| **Simpson et al. (2015)[65]** | Reexamination of four squirrels. AFB-staining (n = 4) PCR *M. lepromatosis* of frozen lesion tissues: pinna, feet, flank 'wart', tail lesion, spleen (n = 1). | Yes | No | Yes | Isle of Wight | Red squirrels | • PCR: 1/1 (*M. lepromatosis*) • AFB: 2/3 |
| | | | | | Brownsea Island | Red squirrel | AFB: 1/1 |
| **Avanzi et al. (2016)[23]** | PCR tested for both *M. leprae* and *M. lepromatosis*. Tissue samples (n = 114) varied per animal in tissue quantity and type. Blood/sera samples for PGL I if available (n = 27). | No | Yes | Yes | England | Red squirrels (n = 26) | • PCR: 26/26 (*M. leprae* n = 25, *M. lepromatosis*, n = 1) • PGLI: 13/22 (59.1%) |
| | | | | | Ireland | Red squirrels (n = 40) | PCR: 3/40 (7.5% *M. lepromatosis*) |
| | | | | | Scotland | Red squirrels (n = 44) | • PCR: 6/44 (13.6%, *M. lepromatosis*) • PGLI: 1/1 |
| | | | | | | Grey squirrels (n = 4) | • PCR: 0/4 • PGLI: 0/4 |
| **Butler et al. (2017)[66]** | Ear pinnae and skin samples (n = 92), PCR for *M. leprae* and *M. lepromatosis*. | No | No | Yes | Isle of Wight | Red squirrels (n = 92) | PCR: 1/92 (1.1%, *M. lepromatosis*) |
| **Schilling et al. (2019)[67]** | PCR for *M. lepromatosis* and *M. leprae*. UK: pinnae, footpads. France: Pinnae. Italy: Pinnae. Germany: Pinnae, footpads. Switzerland: Pinnae. Mexico: Liver, lymph nodes, muzzle, footpad. | No | No | Yes | United Kingdom | Eastern Grey Squirrel (n = 64) | 0/373 |
| | | | | | France | Pallas's squirrels (n = 64) | |
| | | | | | | Siberian chipmunk (n = 35) | |
| | | | | | | Eurasian red squirrels (n = 26) | |
| | | | | | Germany | Eurasian red squirrels (n = 22) | |
| | | | | | Switzerland | Eurasian red squirrels (n = 5) | |
| | | | | | Italy | Eurasian red squirrels (n-43) | |
| | | | | | | Pallas's squirrels (n = 39) | |
| | | | | | | Eastern Grey Squirrel (n = 3) | |
| | | | | | Mexico | White-throated woodrats (n = 72) | |
| **Tió-Coma et al. (2020)[68]** | PCR for *M. lepromatosis* and *M. leprae* spleen samples of squirrels from The Netherlands, spleen and liver samples of Belgian squirrels. | No | No | Yes | The Netherlands | Red squirrel (n = 61) | 0/115 |
| | | | | | | Japanese squirrel (n = 1) | |
| | | | | | Belgium | Red squirrel (n = 53) | |

ancestor around 27,000 years ago. The *M. leprae* strain appeared closest to the one that circulated in medieval England. It is therefore likely that the original introduction of *M. leprae* into red squirrels occurred when leprosy was still prevalent in the region. The medieval squirrel fur

trade might have played a role in the transmission of *M. leprae* between red squirrels and humans.

Red squirrels were found to be a reservoir for *M. leprae* and *M. lepromatosis* in the British Isles, but the prevalence of *M. leprae* infection is variable and often low in the different UK red squirrel populations.[23,67] Between 2013 and 2016, a follow-up study on the Isle of Wight was done to determine the infection rate in red squirrels.[66] Samples from red squirrels (n = 92) found dead were examined by PCR for *M. leprae* and *M. lepromatosis*. One of the squirrels tested positive for *M. lepromatosis*, none for *M. leprae*.

No evidence has been found for infection with *M. leprae* in squirrels or wild rodents outside the UK. Schilling et al. (2019) studied the white-throated woodrat (*Neotoma albigula*, n = 72) from Mexico, and from Europe (England, France, Germany, Switzerland and Italy) the Siberian chipmunk (*Tamias sibiricus*, n = 35), Eurasian red squirrel (n = 96), Eastern gray squirrel (*Sciurus carolinensis* n = 67), and Pallas's squirrel (*Callosciurus erythraeus* n = 103). All samples of all these animals tested with PCR for *M. lepromatosis* and *M. leprae* were negative.[67]

Tió-Coma et al. (2020) examined sixty-one red squirrels and one Japanese squirrel (Sciurus lis) from the Netherlands and fifty-three red squirrels from Flanders, Belgium for the presence of lesions and *M. leprae* and *M. lepromatosis* DNA.[68] No clinical signs of leprosy were observed. All samples were negative in PCR analysis.

**Nonhuman primates.** Cases of nonhuman primates with leprosy have been reported sporadically. All case reports described here involve imported animals. These cases were observed in highly monitored animal research facilities, not in the wild.

In 1977, leprosy-like symptoms were reported in a captive chimpanzee (*Pan troglodytes*) by Donham and Leininger.[69] Follow-up of disease progression and post-mortem examination confirmed that the pathogen was either *M. leprae* or a pathogen indistinguishable from it. [70,71] Hubbard et al. (1991) reported natural infection with *M. leprae* of two chimpanzees. [72] The animals had been in a United States research facility for over 25 years. Gormus et al. (1991) tested sera of these chimpanzees that had been naturally infected with leprosy and of 160 other chimpanzees housed in two primate centers.[73] An ELISA antibody assay for PGL I and the non-specific mycobacterial antigen lipoarabinomannan (LAM) was used. Seven animals were positive for anti-PGLI, and five for anti-LAM antibodies. In 2010, Suzuki et al. reported leprosy in a chimpanzee imported from Sierra Leone used for medical research in Japan.[74] SNP type of the leprosy strain was determined. The strain was SNP type 4, a strain found in West Africa, in line with the origin of the chimpanzee.

In 1979, a sooty mangabey (*Cerocebus atys*) monkey imported into the USA from West-Africa was observed with leprosy-like symptoms in the face.[75] The etiologic agent was identified as *M. leprae* by the following criteria: invasion of nerves, staining properties, electron microscopic findings, non-cultivability, lepromin reactivity, infection patterns in mice and armadillos and sensitivity to sulfone. Supposedly, the animal acquired the disease from a patient with active leprosy.[76] A second captive sooty mangabey monkey with naturally acquired leprosy was reported, that had been in direct contact with the first sooty mangabey monkey, and leprosy was believed to have been transmitted between those two monkeys.[77]

Hagstad (1983) screened 5 rhesus monkeys (*Macaca mulatta*) and 21 bonnet monkeys (*Macaca radiata*) in Andhra Pradesh, India.[78] The monkeys were owned as pets (n = 3) or used for begging (n = 23). The 26 monkeys were in frequent contact with handlers in a leprosy endemic area. No AFB were observed in ear lobe tissue smear slides.

Valverde et al. (1998) reported the first case of leprosy in an Asian macaque (*Macaca fascicularis*) imported from a leprosy endemic area in the Philippines. Diagnosis of infection was obtained by a PCR specific for *M. leprae*. Clinical presentation, histopathological findings, and serology of anti-PGL-I were compatible with human borderline (BB) leprosy.[79]

Honap et al. (2018) sequenced *M. leprae* genomes from frozen tissue samples of 3 of the above infected nonhuman primates (the chimpanzee from Sierra Leone, the first sooty mangabey from West Africa and the cynomolgus macaque from The Philippines).[80] *M. leprae* strains from the chimpanzee and sooty mangabey monkey belong to the human *M. leprae* Branch 4 lineage commonly found in West Africa. Phylogenetic analysis also showed that the cynomolgus macaque *M. leprae* strain belongs to *M. Leprae* Branch 0 and is most closely related to the modern human *M. leprae* strain S9 from New Caledonia (southwestern Pacific). In addition, samples from wild ring-tailed lemurs (*Lemur catta*, n = 41) originating from Madagascar and wild chimpanzees (*Pan troglodytes schweinfurthii*, n = 22) from Kibale National Park, Uganda were examined. DNA was extracted from the samples for detection of *M. leprae* and *M. tuberculosis*, and a variety of mycobacteria pooled with one sequence DNA using qPCR. All samples were negative.

**Screening for new animal species.**   The detection of leprosy infection in new species has, until now, resulted from coincidental observations. Screening of new species has been performed to find potential wildlife reservoirs. Finding *M. leprae* or *M. lepromatosis* would justify further study. The above mentioned studies by Schilling et al. (2019), Hagstad (1983) and Honap et al. (2018) can also be considered screening studies.

Job and colleagues (1988) looked for leprosy infections in small wild animals in Louisiana, in view of the discovered high prevalence in wild armadillos.[81] Both ears from road killed rabbits (n = 51), nutria (n = 56), raccoons (n = 17), and opossums (n = 311) were examined with a modified Fite's stain. All samples were negative.

In the armadillo study by Pedrini et al. (2010), road killed species were examined for *M. leprae* DNA by PCR: ring-tailed coati (*Nasua nasua*, n = 2), skunk (*Didelphis albiventris*, n = 1), hedgehog (*Sphigurrus spinosus*, n = 1), South-American raccoon (*Procyon cancrivorus*, n = 1), restless cavy (*Cavia aperea*, n = 1), greater grison (*Gallictis vittata*, n = 2), and crab-eating fox (*Cerdocyon thous*, n = 2). All samples were negative.

Maruyama et al. (2018) screened various wildlife and captive species in Brazil for the presence of *M. leprae* DNA.[82] Nasal swabs were collected from 69 captive (zoo animals) and free wild animals from the leprosy endemic Mato Grosso and Pantanal regions, independent of clinical signs. *M. leprae* DNA was detected in: Owl monkey (*Aoutus trivirgatus;* n = 1), Capuchin monkey (*Sapajus paella;* n = 2), Lowland tapirs (*Tapirus terrestris;* n = 2, 1 captive), Margay (*Leopardus weidii;* n = 1, captive).

**Leprosy-like disease.**   Leprosy-like disease, attributed to mycobacteria, has been reported in a variety of animals. Phylogenetic analysis compares the genetic sequence, or a fragment thereof, and determines the similarity between two strains. It can be visualized in a family tree, where the arms indicate the temporal distance to a previous common ancestor. Direct genetic comparison of genes can reveal overlap in sequences. This can explain overlap in characteristics of pathogens.

In 1926, a cutaneous mycobacteriosis called skin-tuberculosis was reported in Indonesian water buffaloes.[83] A large series of case studies was published under the disease name Lepra bubalorum by Lobel (1936).[84,85] In 1954, Kraneveld and Roza published the post-Second World War status of the disease. They introduced the name Lepra Bovina, as the disease had been observed in bovine species other than the water buffalo.[86] There had been no progression towards clarification of the pathogen. Similar to leprosy, the microorganism could not be cultured or grown in laboratory animals. Since a last report in 1961, no more cases have been published.[87,88] It is unclear whether the condition still exists in Indonesia. Clinical symptoms and histopathological findings indicating neural involvement were not observed.

*Mycobacterium lepraemurium* causes leprosy-like symptoms in rats. The given name is based on the symptoms, and rather misleading, as it suggests close relation to *M. leprae. M.*

*lepraemurium* is part of the *M. avium*-complex, a mycobacterial sub-group different from *M. leprae*, as was shown by phylogenetic analysis.[89] *M. lepraemurium* is one of the pathogens causing feline leprosy.[90–94] Another feline leprosy causing pathogen was discovered, and named Candidatus "Mycobacterium lepraefelis", and this pathogen is closely related to *M. leprae*.[95] Canine leprosy is a disease associated with members of the *Mycobacterium simiae* clade, and thus unrelated to *M. leprae*.[96,97] A thelitis-causing pathogen in dairy goats and cows appeared to be closely related to M. leprae, and was named M. uberis.[98–100]

## Environmental reservoirs (see Table 4)

The environment is defined as any location outside the host, where *M. leprae* resides before either dying or infecting a new host. Indications for the presence of viable *M. leprae* in the environment can be found from several experimental studies portraying the possibility of *M. leprae* to survive extracellularly under a variety of conditions.[101–105] These studies are shown in Table 4, along with more recent studies. More recent experiments are described in this section.

Lavania et al. (2006) took 18 soil samples in 15 villages near JALMA, the national leprosy research institute in India.[106] Six of these samples were positive for *M. leprae* DNA b*y* PCR. DNA is very stable and can be preserved and detected long after deposition. Later studies incorporated the detection of *M. leprae* specific 16S rRNA. RNA is a copy of a selection of the DNA sequence needed for protein production, and has a very short half-life, often only a few hours. Detection of *M. leprae* specific RNA is therefore an indication of the pathogens' viability. Lavania et al. (2008) collected soil samples in the city of Ghatampur, India.[107] A distinction was made between samples from around houses in leprosy patient areas (n = 40) and non-patient areas (n = 40). 24 samples from the patient area were positive by PCR. 22 of these samples were also positive for 16S rRNA. Six samples of the non-patient area were positive by PCR.; all six were also positive for 16S rRNA.

Turankar et al. (2012) took 207 soil samples of areas where leprosy patients reside in the Purulia district, West Bengal, India.[108] Seventy-one samples were positive for *M. leprae* DNA, and 28 contained viable *M. leprae* based on rRNA PCR. It was attempted to compare patient and environment samples using SNP and VNTR methods, but only SNP type comparison succeeded. SNP type 1 was detected, but subtypes varied among soil samples and patients, even when patients were related. Turankar et al. (2016) repeated their earlier study in the same district.[109] Of 160 soil samples, 52 contained *M. leprae* DNA. *M. leprae* specific RNA was detected in 16 of these positive samples.

Mohanty et al. (2016) took soil and water samples in India from leprosy endemic areas close to patients, and control samples from areas considered leprosy-free.[110] 25.4% of the soil samples and 24.2% of the water samples from the leprosy endemic area were positive for *M. leprae* 16S rRNA, while all the control samples were negative.

Holanda et al. (2017) found *M. leprae* DNA in water sources in five municipalities of Ceará, Brazil.[111] Arraes et al. (2017) continued in these districts and analyzed newly acquired samples for viable leprosy specific *gyrA* mRNA.[112] Samples were taken from environmental water surfaces, of which 76.7% contained *gyrA* mRNA. Tió-Coma et al. (2018) expanded studies of *M. leprae* in soil.[113] Soil samples from 2–8 cm. depth were collected from 3 geographically different areas: Bangladesh, England, and Suriname. In Bangladesh, samples were collected close to leprosy patients' bedrooms. In Suriname, samples were taken from areas inhabited by the nine-banded armadillo: the villages Pikin Slee and Gujaba, and the former leprosy colonies Batavia and Groot Chatillon. In England and Scotland, samples were taken from Brownsea Island (*M. leprae* infected red squirrel area), and the Isle of Arran (*M.*

**Table 4. Environmental reservoirs of *M. leprae* and *M. lepromatosis*.**

| Sample source and type. | Authors (publication year) | Methods | Results |
|---|---|---|---|
| Laboratory: Patient material (biopsy and nose blow). | Desikan (1977) [101] | Purified patient samples dried on Petri-dishes up to 9 days. MFP. | *M. leprae* retained viability tested with mouse footpad method. |
| Environment: Sphagnum from the Norwegian Atlantic coastline. (n = 132). | Kazda and col. (1980)[14] | Sphagnum suspension into MFP. | 29.6% of samples contained acid-fast bacilli, especially in former leprosy endemic regions. |
| Environment: Soil, water and sphagnum samples from nine countries (n = 729): Norway (n = 273), Ivory Coast (n = 71), Portugal (n = 36), Peru (n = 30), India (n = 20), Louisiana (n = 67), Sweden (n = 40), Scotland (n = 77), Germany (n = 115). | Kazda (1981)[105] | Sphagnum suspension into MFP. | • Norway 32.9%<br>• Ivory Coast 23.9%<br>• Portugal 55.6%<br>• Peru 40%<br>• India 30%<br>• Louisiana 25.5%<br>• Sweden, Scotland and Germany 0%. |
| • Environment: Sphagnum from Norwegian Atlantic coastline.<br>• Laboratory: Armadillo derived *M. leprae* into sphagnum. | Kazda et al. (1990) [114] | Sphagnum suspension into MFP and detection with anti-PGL-I antibodies. | MFP grown sphagnum isolates contained PGL-I, successful laboratory replication. |
| Laboratory, patient material; fresh leproma samples. | Kaur et al. (1982) [102] | • 9d dried samples (Desikan,1977).<br>• 30 minutes of irradiation.<br>• Two hours of direct sunlight.<br>• Seven days at room temperature. | All tests: *M. leprae* retained viability as tested with MFP. |
| Laboratory, patient material; skin biopsy. | Desikan en Sreevatsa (1995) [103] | Mouse footpad inoculation, and cell counting. Relative humidity (RH), Temperature range (T). | • RH: 72–80%, T: 29–33˚C (monsoon), 14 days<br>• RH: 28–44%, T: 24–33˚C, 28 days.<br>• RH: 30–40%, T: 38–40˚C (3 hours of direct sunlight a day), 7 days.<br>• Wet soil, 46 days.<br>• T: 4˚C, -20˚C, and -70˚C: 60 days, 60 days, and 28 days respectively. |
| Soil samples near leprosy residences. (n = 18) | Lavania et al. (2006)[106] | PCR samples: RLEP *M. leprae* specific region. | 6/18 (33.3%) positive for RLEP region. |
| Soil samples from leprosy residences. | Turankar et al. (2012)[108] | *M. leprae* specific DNA and *M. leprae* specific 16S rRNA. | • DNA: 71/207 (34.3%)<br>• rRNA: 28/71 (39.4%) |
| Soil samples from leprosy residences. | Turunkar et al. (2016)[109] | *M. leprae* specific DNA and *M. leprae* specific 16S rRNA | • DNA: 52/160 (32.5%)<br>• rRNA: 16/52 (30.8%) |
| Soil and water samples from endemic leprosy area and leprosy free area (no cases for 5–6 years). | Mohanty et al. (2016)[110] | *M. leprae* specific 16S rRNA. | • Soil: 43/169 (25.4%)<br>• Water: 41/169 (24.2%) |
| Five natural water sources. Water samples from surface and per 25cm until 100cm depth, several sites per water source. | Holanda et al. (2017)[111] | *M. leprae* specific *gyrA* mRNA. | *M. leprae* specific mRNA in 23 (76.7%) of the water sources. |
| Soil samples from 2–8 cm. depth. | Tió-Coma et al. (2019)[113] | *M. leprae* specific DNA and *M. lepromatosis* specific DNA. | *M. leprae* specific DNA was detected: Bangladesh: 16% (4/21), Suriname: 10.7% (3/25), Brownsea Island: 10% (1/10), Isle of Arran: 0/10. No samples contained *M. lepromatosis* specific DNA. |

*lepromatosis* infected red squirrel area). *M. leprae* specific DNA was found in samples from Brownsea Island, Suriname, and Bangladesh, but not the Isle of Arran. *M. lepromatosis* DNA could not be detected in soil samples from the Isle of Arran.[114]

Turankar et al. (2019) studied the association between non-tuberculous mycobacteria and *M. leprae* in the environment of leprosy endemic regions in India. They analysed soil (n = 388) and water (n = 250) samples for RLEP DNA and 16S rRNA. RLEP DNA was detected in 118 soil samples (30%) and 48 water samples (19%). 16S rRNA was detected in 53 soil samples (14%) and 30 water samples (12%).[115]

## Zoonotic and anthroponotic transmission

Filice et al. (1977) tried to assess, in a limited study, whether armadillo exposure is a risk factor for developing leprosy. Leprosy patients in Louisiana (n = 19) did not have more exposure to armadillos than neighborhood matched controls (n = 19).[116]

Deps et al. (2008) carried out the same type of study in Brazilian leprosy patients and a control group of patients with chronic diseases other than leprosy.[117] Exposure to armadillos was categorized as non-existent, indirect or direct. Physical contact was considered direct contact and living in an armadillo habitat was considered indirect contact. Direct exposure was significantly higher in leprosy patients (68%, n = 506) than in the control group (48%, n = 594) with OR 2.01 (95%CI 1.36–2.99, $p$ = 0.0001).

Schmitt et al. (2010) studied the relation between armadillo meat consumption and leprosy in a case-control study of Brazilian leprosy patients (n = 121) and patients with other skin diseases (n = 242).[118] No significant difference was observed in armadillo meat consumption between the two groups: OR 0.77–1.90 ($p$ = 0.44). The study supported, however, an association with unfavourable socioeconomic indicators as leprosy patients differed significantly ($p$<0.05) from controls in access to treated water (90% vs. 96%), lower family income based on minimum wage, and more contact with other leprosy patients (35% vs. 7%)

Truman et al. (2011) used whole-genome sequencing of *M. leprae* on isolated DNA from human skin biopsies of 50 American leprosy patients and armadillo liver, spleen or lymph-nodes of 33 Louisiana captured nine-banded armadillos.[119] Seven SNP subtypes were found in all samples, but selection of patients without travel history (n = 29) reduced the number of SNP subtypes to four. SNP subtype 3I, generally associated with European-American populations was present in all armadillos and 26 of 29 American patients without travel history. Further analysis of subtypes using variable number tandem repeat (VNTR) profiles showed that one specific *M. leprae* genotype (3I-2-v1) was present in 28 of 33 wild armadillos (85%) and 25 of 39 patients (64%) who resided in areas where exposure to armadillos was possible. They concluded that human leprosy in Louisiana and Texas may result from contact with infected armadillos.

Sharma et al. (2015) tested for *M. leprae* subtypes in the southeastern USA using VNTR SNP (sub-)type identification.[19] They found serologic and PCR evidence for *M. leprae* infection in 106 (16.4%, n = 645) nine-banded armadillos from eight locations in Alabama, Florida, Georgia, and Mississippi. Human *M. leprae* was isolated from frozen skin biopsies of 52 leprosy patients from the same geographic region as the sampled armadillos. Lymph node tissue for PCR analysis was obtained from 95 of the 106 infected armadillos. 42 of 95 samples contained enough DNA for genotyping. SNP subtype 3I-2-v1 was present in 35 of 42 armadillos (83%) and 12 of 52 humans (23%). The newly identified SNP subtype 3I-2-v15 was found in 7 of 42 armadillos (17%) and 10 of 52 humans (19%). SNP subtype 3I-2-v15 has not been reported outside the state of Florida. The leprosy patients with this genotype all lived in Florida. These data show that the *M. leprae* strains currently circulating in this part of the USA are closely related to medieval European *M. leprae* strains and suggest that leprosy was brought to the southern United States by the early European settlers since the sixteenth century. Armadillos must have acquired *M. leprae* from humans in the past 400 years.

Da Silva et al. (2018) have reportedly found evidence for zoonotic leprosy in Brazil, but actually studied leprosy occurrence in armadillos, and armadillo related risk factors for human leprosy and anti-PGL I titers separately.[120] They determined infection in 10/16 armadillos (62%) by PCR of liver and spleen samples. Blood samples from the local population (n = 146) were analyzed for PGL I. Stratification was performed based on hunting or eating armadillos, and analyzed further based on frequency. Of 146 individuals, over the previous year, 27

(18.5%) hunted armadillos, 96 (65.8%) handled or prepared the meat for consumption, 91 (62.3%) ate armadillo meat at least once, and 27 (18.5%) ate them more than once per month. The number of leprosy patients among armadillo hunters (4 of 27, 14.8%) was significantly higher than among non-hunters (3 of 119, 1.8%) OR 6.73 (95%CI 1.41–32.09 p = 0.02). Anti-PGL I titers were different when stratifying for exposure frequency. Eating armadillo meat more than 12 times per year was associated with an insignificant increased risk of PGL I positive antibody titers (OR 1.77 95%CI 0.64–4.89). The median anti-PGL I titer was significantly increased in this group (n = 27), compared to never eating (n = 55, *p* = 0,03) and eating 1–12 times per year (n = 64, *p* = 0.01).

Stefani et al. (2019) studied nine-banded armadillos (n = 12) and local volunteers (n = 176) in the rural endemic area of Mamiá Lake of the Coari municipality, Brazil.[121] The armadillos were supplied locally. Volunteers were examined for clinical signs of leprosy by a dermatologist. Armadillo skin, spleen, liver, lymph, adrenal glands, ovary and fallopian tubes and human skin lesions suspect of leprosy were biopsied and examined histopathologically. All tissues from armadillos were tested with qPCR for the *M. leprae* specific RLEP sequence. Six new leprosy patients were identified among the local volunteers. None of the armadillo samples showed *M. leprae* DNA or AFB.

## Potential vectors (see Table 5)

Attempts at discovering vectors for *M. leprae* have been reported throughout the twentieth century. In older studies, leprosy bacilli viability could not be shown by culture or the use of animal models. These studies are not discussed in depth, because of their limited impact. Three studies contained only a description of the theoretical role of arthropods in the transmission of leprosy.[122–124] In several studies, leprosy patients were used to test the uptake of leprosy bacilli from lesions.[125–131] In others, leprous material from patients was used to feed insects.[132–134] Insects were also collected from leprosy patient's homes.[131,135–137] None of these studies were able to show possible transmission by insects. At most, insects would take up the bacteria or transmit a small number of bacilli to a subsequent feeding target, but this would not be considered a valid transmission route.

*M. leprae* is an obligate intracellular pathogen. Intracellular microorganisms have been associated with free living amoebae (FLA) without a known benefit for either. It is currently proposed to define these microorganisms and FLA as "endocytobionts". Such a relationship was studied for *M. leprae*. Phagocytosis of fluorescent *M. leprae* by amoebae was observed by Lahiri and Krahenbuhl (2008) and Wheat et al. (2014). Inside the amoebae *M. leprae* was seen flocking to the more acidic regions of the cell.[138,139] Lahiri and Krahenbuhl (2008) showed that *M. leprae* remained viable for at least three days as assessed by MFP.[138] Wheat et al. (2014) found that *M. leprae* remained viable for 35 days in *Acanthamoebae lenticulata*, *A. castellanii*, *A. polyphaga*, *Hartmannella vermiformis*, and *H. vermiformis*, and up to 8 months within amoebic cysts.[139] The degree of *M. leprae* phagocytosis by *Acanthamoeba sp*.S-11 was shown by Paling et al. (2018) to be nutrition driven.[140] Amoebae would phagocytose co-cultured *M. leprae* increasingly in poor nutrient conditions.

As amoebae are present worldwide in the environment (e.g., soil, water and air), they may potentially play a role in the survival of *M. leprae* outside the mammalian host. Exposure of mammals to infected amoebae could be a risk factor for transmission. In this form of transmission, the amoebae would function as a vector. Turankar et al. (2019) found an association between *M. leprae* DNA and *Acanthamoeba* DNA in environmental samples.[115] Viable *M. leprae* could be found using 16S rRNA detection in 106 of 700 soil samples (15%), and 34 of 400 water samples (8%). *Acanthamoeba* DNA was associated with viable *M. leprae*. Of the 106

**Table 5. Potential vectors of *M. leprae*.**

| Vector | Authors (publication year) | Analysis | Goal | Results |
|---|---|---|---|---|
| Amoeba (*Acanthamoeba castellanii*) | Lahiri and Krahenbuhl (2008)[138] | Viability tested with AFB staining for intact cell wall, metabolic activity of *M. leprae* by radiospirometry, and phagocytosis isolates into MFP. | Test viability of *M. leprae* after phagocytosis by *A. castellanii*. | Viable 3 days after phagocytosis. |
| Amoebae (*A. castellanii* and *A. polyphaga*) | Wheat et al. (2014)[139] | Viability tested with MFP. | Test viability of *M. leprae* after phagocytosis and amoebic cysts in five different amoebae. | Viable for over 30 days after phagocytosis. Viable up to 8 months in amoebic cysts. |
| Mosquitoes (*Aedes aegypti* and *Culex quinquefasciatus*) and Kissing bugs (*Rhodnius prolixus*). | Neumann et al. (2016)[141] | Viability tested with PCR 16S rRNA analysis of gut samples. MFP of *R. prolixus* feces samples. | Test viability in vector and transmission potential. | Mosquitoes did not maintain viability of *M. leprae*. Kissing bug gut passage and kissing bug feces contained viable *M. leprae*. |
| Female ticks, their eggs and larvae (*Amblyomma sculptum*) | Da-Silva Ferreira et al. (2018)[142] | Viability tested with PCR 16S rRNA from gut and ovaria samples. qPCR analysis of rabbit skin samples. | Viability of *M. leprae* in female ticks. Transmission of larvae to rabbit skin. | Viable *M. leprae* in ovaria and gut. $10^3$ viable *M. leprae* in rabbit skin samples |

soil samples positive for 16S rRNA, 30 were also positive for *Acanthamoeba* DNA (28%). Of the 112 negative soil samples, 10 were positive for *Acanthamoeba* DNA (9%). Water samples showed the same association. *Acanthamoeba* DNA was detected in 14 out of 34 16S rRNA positive samples (41%) and 1 out of 39 *M. leprae* DNA positive 16S rRNA negative soil samples (2.6%). The association between viable *M. leprae* and *Acanthamoeba* DNA was significant for soil (p = 0.002) and water (p = 0.009).

Neumann et al. (2016) investigated the possibility of transmission of *M. leprae* by the kissing bug (*Rhodnius prolixus*) and mosquitoes (*Aedes aegypti* and *Culex quinquefasciatus*).[141] Both are known to draw blood from armadillos. The mosquitoes, however, did not meet the essential criterion for a vector: the ability to maintain an infectious load of the pathogen until the next feed. But *M. leprae* remained viable in kissing bugs for 20 days. Inoculation of MFP with feces of kissing bugs proved that *M. leprae* had remained viable. This is a well-known transmission pathway for *Trypanosoma cruzi*, causing the zoonotic, vector transmitted Chagas disease. *T. cruzi* remains viable during the kissing bug's gastrointestinal tract passage. Kissing bugs transmit disease, not by biting, but by defecating near the bite wound. Rubbing or scratching infects the wound.

Da Silva Ferreira et al. (2018) achieved similar results with the adult female tick (*Amblyomma sculptum*).[142] Ticks were fed on rabbit blood containing *M. leprae*. Detection of *M. leprae* RNA in the midgut and ovaries of the ticks by qPCR analysis proved viability of the *M. leprae* until at least 2 days after feeding. *M. leprae* was visualized in gut samples with LAM-antibody immune-localization. In addition, *M. leprae* would, in some cases, spread to the ovaries of female ticks and subsequently infect tick eggs and larvae. Infected larvae were attached to rabbit skin for a five day maturation feeding period. Afterwards, up to $10^3$ viable *M. leprae* could be isolated from the skin samples around the bite-mark. The study showed an ecological cycle of *M. leprae* in an experimental setting. The research group also performed the first successful cell culture of *M. leprae* in tick-IDE8 cells.

## Discussion

This systematic review assessed publications of possible non-human environmental reservoirs and transmission pathways of *M. leprae* and *M. lepromatosis*. The results show a wildlife reservoir of *M. leprae* in armadillo species in the southern United States and a wildlife reservoir of

*M. leprae* and *M. lepromatosis* in British and Irish red squirrels. The diagnosis of leprosy in captive non-human primates is limited to case-reports, and it is unknown to what degree species are infected. Screening for the presence of these mycobacteriae in other species has been negative, except for the finding in 2018 of genetic material of *M. leprae* in nasal swabs from a variety of animal species in Brazil.

Possible routes of transmission are schematically visualized in Fig 2. It is justified to ask whether or not we can speak of an environmental reservoir. RNA indicating the presence of viable *M. leprae* was found in environmental soil and water samples in Brazil and India.[143] *M. lepromatosis* has not yet been found in such samples, but this pathogen has only been discovered recently. In addition, it was found that amoebae are capable of taking up *M. leprae* by phagocytosis. Inside amoebae, *M. leprae* remained viable for days to weeks. This mechanism might contribute to environmental survival in the absence of a mammalian host. Moreover, *M. leprae* can also accumulate in amoebae. This gives amoebae a possible role as a vector in transmission. The only studies on vegetation reservoirs were on sphagnum species by Kazda (1980) (Table 4).[103,104] There are no recent studies using PCR techniques on the role of vegetation as an environmental reservoir.

Humans and armadillos in the Southern United States share a specific *M. leprae* strain (SNP subtype 3I-2-v15). This finding is highly suggestive of a zoonotic and/or an anthroponotic transmission pathway between humans and armadillos. Exposure of humans to armadillos can eventually lead to leprosy and/or increased anti-PGL I antibody levels. Exposure to an animal or to animals' excreta is influenced by behavior. For example, hunting and/or preparing armadillos for eating can each be expected to have a different risk of transmission, based on differences in exposure intensity and frequency. It is unclear whether transmission risk by rate of exposure to infected mammals is confounded by rate of exposure to an infected environment. This could be caused by shedding of leprosy bacteria by infected mammals.

At this moment, there is no evidence for a role for vectors in transmission. However, recent laboratory studies have shown a potential role for insects. *M. leprae* remains viable in the gastrointestinal tract of kissing bugs and is experimentally transmittable to the MFP through their feces, similarly to the transmission mechanism of *T. cruzi*. Kissing bugs can defecate after biting, and defecation near the bitten area transmits viable *T. cruzi* when the itchy bitten area is scratched, and feces is rubbed into the wound. In the same study, it was found that *M. leprae* does not remain viable within the gut of mosquitoes. It also has been shown that *M. leprae* can remain viable and reproduce in ticks, and spread to tick ovaria and eggs. Larvae from hatched eggs in turn are able to transmit viable *M. leprae* into a host during skin-bound maturation.

Animals can be affected by leprosy-like diseases, caused by pathogens phylogenetically closely related to *M. leprae*. Feline leprosy is caused by several pathogens, of which Candidatus "*Mycobacterium lepraefelis*" is closely related to *M. leprae*. *M. uberis*, the causative agent of nodular thelitis, is also closely related to *M. leprae*. *M. haemophilum*, which has been reported to infect the skin of immunocompromised human patients, was also found to be closely related to *M. leprae*.[144] The pathogen causing a mycobacteriosis called Lepra Bubalorum in Indonesian water buffaloes could also be related to *M. leprae*, but further research is needed to confirm this. These mycobacteria have been proposed to be grouped as a *M. leprae*-complex.[145] Such a classification connects the related mycobacteria, similar to species of the *M. avium*-complex. Insights from the transmission and reservoirs of members of the *M. leprae*-complex could be relevant for other pathogens in the complex. Currently, knowledge on the exact transmission mechanisms of *M. leprae* is poor. Using insights on related pathogens is a potentially efficient way to progress.

The studies included in this review differ in methods to determine the causative microorganism of leprosy. Techniques have developed over time from identifying unspecified non-

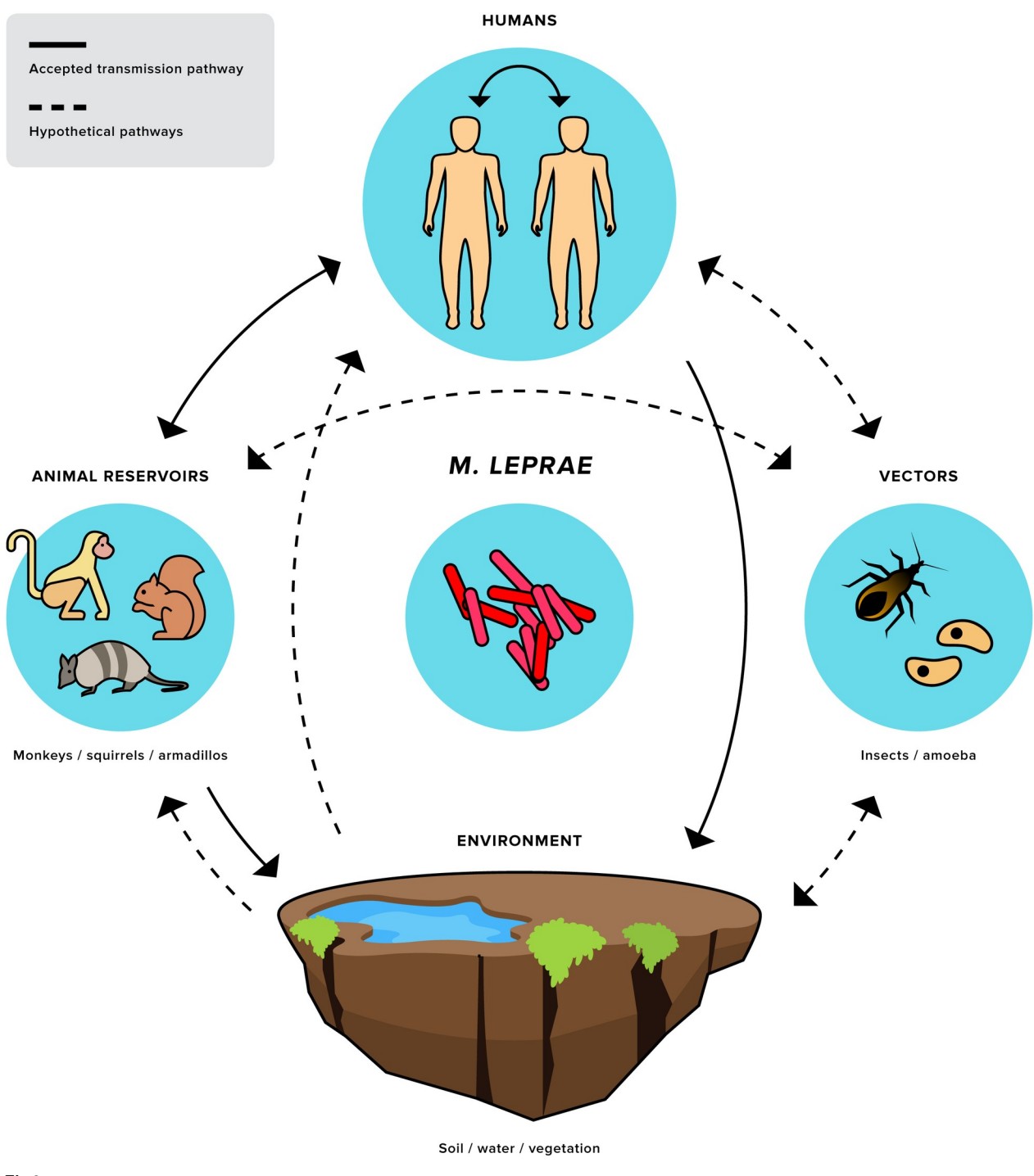

**Fig 2.**

cultivable acid-fast bacteria to specific DNA sequencing. The ability to detect *M. leprae* DNA with PCR techniques drastically changed the methods of modern studies. The variety of and changes in analysis methods for detection of *M. leprae* have resulted in poor standardization of methods, each with their particular strengths and weaknesses. This is also relevant for studies on *M.leprae* and *M.lepromatosis* in epidemiological settings. Anti-PGL I antibody detection in

blood or sera indicates past or present exposure to *M. leprae* but provides little information on current infection status. Studies on zoonosis and prevalence should be performed with sample sizes determined by the population sizes. Results of these studies have to be reported with confidence intervals, as this would substantiate the extrapolation value of the studies. Many studies in this review lack this method of sample size determination. Geographic origins of the animals are described clearly in most armadillo prevalence studies. It is seen that prevalence fluctuates strongly between neighboring geographic regions. This limits the value of spatially and temporally randomly acquired (frozen) samples in screening studies, which often do not represent a wildlife (sub)population. Seldom are internal control standards like those normally used in clinical laboratories incorporated. This lack of rigor contributes to the anecdotal assessment sometimes given to molecular studies reporting *M. leprae* in non-human sources. *M.leprae* rRNA has been used as a viability marker in environmental studies.[108–111] Accuracy of this method was suggested to be inadequate.[146] A reliable standard for viability is needed. Standardization in environmental studies should be increased by first developing appropriate definitions and sample criteria.

This systematic review underscores that human-to-human transmission is not the only way leprosy can be acquired. The transmission of this disease is probably much more complicated than was thought before, as indicated in Fig 2. Transmission of *M. leprae* involves several factors and pathways. Fig 2 shows the animal and environmental factors that might play a role in the persisting prevalence of leprosy. The proposed factors and mechanisms transgress the disciplines of human healthcare. A reduction in transmission cannot be expected to be accomplished by actions or interventions from the human healthcare domain alone, as the mechanisms involved are complex. Therefore, to increase our understanding of the intricate picture of leprosy transmission, a *One Health* transdisciplinary research approach is required. [138] This entails integrating human, animal, and environmental health aspects to further elucidate the transmission mechanisms and patterns of *M. leprae* and *M. lepromatosis*. In addition, geographically tailored methods–combining epidemiological, laboratory, and anthropologic data–are needed to better understand the ecological differences between leprosy pockets.

## Supporting information

**S1 PRISMA Checklist.**
(DOC)

**S1 Text.**
(DOCX)

## Acknowledgments

The authors would like to thank Marja Kik for her feedback on an earlier version of this paper and Toon van der Gronde for his feedback on the methodological section. Furthermore, we would like to thank Nathalie Kuijpers for her English manuscript correction services. In addition, we would like to thank Frank-Jan van Lunteren for his comprehensive graphics (See, Fig 2).

## Author Contributions

**Conceptualization:** Thomas Ploemacher, William R. Faber, Henk Menke, Toine Pieters.

**Data curation:** Thomas Ploemacher.

**Formal analysis:** Thomas Ploemacher, Toine Pieters.

**Investigation:** Thomas Ploemacher, William R. Faber, Henk Menke, Toine Pieters.

**Methodology:** Thomas Ploemacher, William R. Faber, Toine Pieters.

**Project administration:** Toine Pieters.

**Resources:** Toine Pieters.

**Supervision:** Toine Pieters.

**Writing – original draft:** Thomas Ploemacher, William R. Faber, Henk Menke, Toine Pieters.

**Writing – review & editing:** William R. Faber, Henk Menke, Victor Rutten, Toine Pieters.

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
