## [Decision Letter · Decision Letter 0]

7 Feb 2020

Dear Dr. Pieters,

Thank you very much for submitting your manuscript "Reservoirs and Transmission Routes of Leprosy;

A Systematic Review" for consideration at PLOS Neglected Tropical Diseases. As with all papers reviewed by the journal, your manuscript was reviewed by members of the editorial board and by several independent reviewers. In light of the reviews (below this email), we would like to invite the resubmission of a significantly-revised version that takes into account the reviewers' comments. 

We cannot make any decision about publication until we have seen the revised manuscript and your response to the reviewers' comments. Your revised manuscript is also likely to be sent to reviewers for further evaluation.

Sincerely,

Carlos Franco-Paredes

Associate Editor

Ana LTO Nascimento

Deputy Editor

Reviewer's Responses to Questions

**Key Review Criteria Required for Acceptance?**

**Methods**

-Are the objectives of the study clearly articulated with a clear testable hypothesis stated?

-Is the study design appropriate to address the stated objectives?

-Is the population clearly described and appropriate for the hypothesis being tested?

-Is the sample size sufficient to ensure adequate power to address the hypothesis being tested?

-Were correct statistical analysis used to support conclusions?

-Are there concerns about ethical or regulatory requirements being met?

Reviewer #1: This is a review article that utilises a standard literature review PRISMA approach which is appropriate. The objective of providing an overview of worldwise leprosy research regarding reservoirs and transmission is stated at the end of the introduction. The search criteria and inclusion /exclusion process are adequately described, as is how the data were analysed. 

p6 it is unclear to me what the meaning and significance of the sentence " No review protocol was drafted nor registered in advance of this study" is, please clarify.

PRISMA acronym definition and reference should be included in the text p6

p6 It is unclear where the additional 57 articles found through snowballing are included in Figure 1, from the text it reads as if there are 80 +57 but this is not reflected clearly in Figure 1

 " A discussion of the quailty of methods has been incorporated" needs further qualification, as this was not done for all studies and how'quality' is asssessed is unclear. Suggest reword along the lines of ' discussion of the limitations of some studies is incorporated where these could have influenced the findngs or conclusions made....' 

There are no ethical or regulatory concerns

Reviewer #2: The authors present here a scholarly meta-analysis of the literature concerning environmental sources of leprosy bacilli. They are uniquely qualified senior investigators with well recognized expertise in medicine, leprosy, medical history as well as veterinary medical sciences. 

Widespread availability of effective drug therapy has markedly diminished the prevalence of leprosy over that last 30 years; but the new case detection rate has remained low and steady over the last decade. Though traditionally considered a disease transmitted between humans, long-term steady prevalence raises the likelihood that sources other than infected humans are involved in leprosy transmission and play important roles in perpetuating the modern infection. To better understand this issue the authors have made a systematic review of the literature concerning environmental sources of leprosy bacilli. They clearly describe their approach to gather and review the literature, which included thousands of pre-selected articles, and their strategy to organize and focus their review on select papers. This is the most thorough review of environmental sources of leprosy bacilli in the last 40 years.

Reviewer #3: (No Response)

**Results**

-Does the analysis presented match the analysis plan?

-Are the results clearly and completely presented?

-Are the figures (Tables, Images) of sufficient quality for clarity?

Reviewer #1: As a review article, the results comprise summaries and some synthesis of published studies, as staetd in the objective

The Figures and tables are clear and useful/essential

Reviewer #2: The results of their literature searches are thoroughly documented and sorted to different topic areas. The 2 accompanying figures clearly detail their process and the scope of their review.

Reviewer #3: (No Response)

**Conclusions**

-Are the conclusions supported by the data presented?

-Are the limitations of analysis clearly described?

-Do the authors discuss how these data can be helpful to advance our understanding of the topic under study?

-Is public health relevance addressed?

Reviewer #1: Here and in the discussion it is not made clear to the reader whether the suggestion of grouping pathogens causing leprosy-like diseases into an M.Leprae-related complex in research and cdocumentatio is proposed and recommended for the first time here by the authors or whether this reflects this usage previous research and publications; if so this should be clearly stated as a new proposal.

Reviewer #2: The authors conclude that the literature is replete with examples of leprosy bacilli being recovered from the environment and that the body of knowledge concerning this issue is expanding in such a manner as the topic can no longer be considered anecdotal. They offer that further reduction in leprosy transmission cannot be expected from the human healthcare domain alone, as the disease cycle and its ecology is likely far too complex. A better understanding of the possible roles of animal or environmental reservoirs is needed. To achieve this, they suggest a paradigm shifting a One Health transdisciplinary research approach incorporating the skills of scientists beyond those traditionally involved in leprosy research, and further integrating leprosy with other neglected tropical diseases.

Reviewer #3: (No Response)

**Editorial and Data Presentation Modifications?**

Reviewer #1: p5 Introduction

 last paragraph states that it is a review regarding 'reservoirs and transmission' but it does not actually review direct human to human transmission and so this should be made clearer.

Synopsis.M.leprae and M. lepromatosis both target skin and peripheral nervous systems, not just M.leprae. 

 line 6 insert 'created by' in front of 'coughing and sneezing..'

 Abstract:

p2 line 3 M.leprae and M. lepromatosis both target skin and peripheral nervous systems, not just M.leprae

 line 5. ' selective prevalence' is not the correct term in my opinion to describe local/geographocal differences in prevlence, as it implies the pathogen 'selects' where it causes disease. An alternative term such as 'spatial inequaility' could be used to describe the endemic pockets.

line 11 insert 'created by' in front of 'coughing and sneezing..'

line 18, you state the two bacilli being 'found' in red squirresl - in fact the DNA of these two bacilli was detected - and it should also be stated that the disease leprosy was also found in this species

line 28. Suggest change the phrase ( and in the conclusion where you use this same wording)" Reducing transmission cannot be expected from the human healthcare domain alone..... to something like " A reduction in transmission cannot be expected to be accomplished by actions or interventions from the human healthcare domain alone,..." 

Introduction

p4 line 6: insert 'created by' in front of 'coughing and sneezing..'

p4 end of 2nd pargrah' ' in cases of leprosy'

p 5 line 1 see comment above re the term selective prevalence

p 5 third paragraph add that the disease leprosy was also found in red squirrels as well as the bacterial DNA detected

Method

p 6 penultimate line, 'Data..... were' ( not was)

Results

p8 line 1 change ' symptoms' to ' clinical signs' - animals cannot have symptoms, only humans, it is not a correct veterinary term

p8 second paragraph 2nd sentence Do not start a sentence with ' But also...'

p16 'Siberian chipmuck ( Tamias sibiricus n=35) ' is duplicated

p17 some common species names are wrong, although I have not checked the original text it would be more useful to give the more usual English common name; Procyon cancrivoras is the crab-eating or South American raccoon, not sure what' hand-skinned' means?. Gallictis vittata is the greater grison, not ferret; Cerdyon thous is the crab-eating fox.

Discussion

p27 line 4 change "English red squiirels ' to British and Irish re squirrels' and see my abocve comments regarding resrevoirs definition - as you do state in the abstract it should be stated throughout that animals and enviroment are possible reservoirs only.

Reviewer #2: Editorial comment suggestions:

An important area not mentioned is that clarifying the importance of environmental reservoirs can also have important direct impact on patients. The stigma and anxiety they experience after a diagnosis of leprosy is often exacerbated by their lack of knowledge about how or from where they may have acquired this infection. Understanding that leprosy has clear environmental or biological origins can be quite beneficial in ameliorating the social and psychological impacts of leprosy. 

In modern times much of the literature includes PCR based assays, usually detecting single target segments of M. leprae or M. lepromatosis DNA. Few investigators perform blinded analyses and even fewer include blinded randomized controls in their assay runs. Seldom are internal control standards like those normally used in clinical laboratories incorporated into studies performed in research laboratories. This lack of rigor contributes to the anecdotal assessment sometimes given to molecular studies reporting M. leprae in non-human sources. The authors should acknowledge the lack of standard controls in many studies and encourage others to exercise greater rigor in environmental studies of Ml and MLPM. 

Similarly, multiple authors suggest that mere detection of M. leprae 16s RNA is indicative of viability of the organisms. However, 16s is produced in massive quantities in bacilli and may have a half-life as long as DNA itself. I am not aware of any literature assessing what level of 16s detection is actually associable with viability of M. leprae -- 16s concentration can be seen to rise and fall during growth or decline of leprosy bacilli, but static detection of 16s is no more meaningful than detecting DNA. Some comment about the lack of a good standard for environmental viability would be useful to include. 

Minor items:

The methods section refers to papers “excluded”. The Supplementary data section refers to papers “rejected”. Please make consistent and add an explanation to the lists in the supplementary sections.

Page 11 last paragraph line 12: I am pretty sure you mean to say “did NOT grow on Lowenstein-Hensen.

Page 22 next to last line: should be …settlers ‘since’ the 16th century.

Page 24: Please comment on viability as above

Reviewer #3: (No Response)

**Summary and General Comments**

Reviewer #1: This is a useful systematic review article providing an overview of global leprosy research , using historical and new research data on leprosy transmission to summarise the current state of knowledge on possible reservoirs and possible transmission routes, to highlight wildlife and environmental reservoirs ( including the recently discovered red squirrel) and encouraging a One Health approach in light of this to future leprosy resreach and interventions. 

It is generally well written but it is clear that English is not first language of the authors and there are numerous grammatical errors and some awkward phrasing throughout, some of which I have specified , eg ' data' is plural not singular - but a further thorough review by a native english speaker is recommended. 

The use of the term reservoir is made throughout. I would recommed a definition of this term is given and some discussion of this integrated, as this is an important concept of great relevance to this manuscript. Recommed the definition by Viana et al (2014): A 'reservoir of infection' is defined with respect to a target population as 'one or more epidemiologically connected populations or environments in which a pathogen can be permanently maintained and from which infection is transmitted to the target population' https://www.ncbi.nlm.nih.gov/pmc/articles/PMC4007595/

For the animal species discussed in this reviewe there is largely only speculative or indirect evidence only of permanent maintenance of the pathogen

Reviewer #2: This is a well written and constructed review by senior scholars with vast expertise in this area. It is likely to become a standard reference for both professionals and students. The topic is important to leprologists and others interested in neglected tropical diseases, and may help bring a better understanding about this disease for medical professionals and patients.

Reviewer #3: The manuscript by Ploemacher, et al., is a review of the current literature on confirmed and possible reservoirs of leprosy bacilli and routes of transmission of leprosy. Although multidrug therapy has reduced the prevalence of leprosy, the number of new cases each year has remained relatively constant. Therefore, the authors argue, human-to-human transmission is not the only way leprosy can be acquired. Animal and environmental reservoirs, as well as vectors, may play a role in transmission of leprosy to humans. The choice of articles included in the review is justified and appropriate. The manuscript is well written and interesting to read.

1. I agree that leprosy transmission is likely much more complicated than we think it is. This is an important area for research. I like the One Health approach.

2. page 4, although the Ziehl-Neelsen stain is of historical value, the authors should mention the Fite stain as it was a vast improvement for detecting M. leprae in tissues.

3. page 4, regarding phylogenetics, there are currently “…at least five BRANCHES (0-4)” but only 4 SNP types.

4. page 11, regarding the study by Amezcua et al., “Homogentates of infected nodules and lymph nodes did NOT grow on Lowenstein-Jensen…”

5. page 13, line 3 should be “Colombia”

6. page 17, second paragraph: should that be SNP type 3K, rather than 3I? 3K is branch 0

7. page 17, regarding the study by Pedrini, et al., “…hand-skinned ‘what’…”? Raccoon?

8. page 19, the authors should provide references for their comment regarding the short half-life of RNA and the appropriateness of using RNA expression as an indicator of viability, especially the use of ribosomal RNA. Messenger RNA has a short half-life and can indicate viability, but the use of ribosomal RNA as a viability indicator is controversial.

9. page 22, regarding the studies by Truman, et al., the concluding sentence is rather strong. They concluded that both armadillos and humans in the area were infected with the same strain of M. leprae and that leprosy may be a zoonosis in the region.

10. page 24, last sentence of first paragraph: “…the fluorescent staining of living bacteria allowed detection of actual bacteria inside the vector” is not clear. Are you stating that only live bacteria fluoresced? Please provide the reference.

11. Tables 1, 2, and 3: The authors present an overview of the methods applied in these papers. One such method is acid fast bacilli (AFB) staining. AFB in the skin, even in a granuloma, is not confirmatory of leprosy; however, AFB in a nerve is. It would greatly improve the manuscript and be much more informative to the reader if the Tables noted whether or not AFB were observed in a nerve, especially in the older papers before PCR was available. This would also help us understand what kind of reservoir an animal may be, e.g., they actually get leprosy disease or they are carriers and shed it into the environment, etc. It would bolster your One Health approach. Also, growth in mouse footpad is not really informative unless it was confirmed to be M. leprae, so that method could be removed.

12. page 27, last paragraph regarding the designation of “an M. leprae-related complex in research and documentation.” I think this is too preliminary; however it will generate discussion. Therefore, if the authors want to make this proposal, they should elaborate and expand their reasons in this Discussion. The important question is how does one define a “complex?” Is it based simply on genetic similarities found through sequencing a few genes? Or, should it be based on disease manifestations? Both? As stated above, mycobacterial infection of the skin, even in granulomas, is not necessarily leprosy-like. What makes leprosy the devastating disease that it is is the fact that it invades and damages nerves.

PLOS authors have the option to publish the peer review history of their article (what does this mean?). If published, this will include your full peer review and any attached files.

Reviewer #1: Yes: Anna Meredith

Reviewer #2: Yes: Richard W. Truman PhD

Reviewer #3: No

Additional comment CFP (Associate Editor): Recommend reviewing article: 

Unsolved matters in leprosy: a descriptive review and call for further research. Franco-Paredes C, Rodriguez-Morales AJ. Ann Clin Microbiol Antimicrob. 2016 May 21;15(1):33. doi: 10.1186/s12941-016-0149-x. PMID: 27209077. This article of which I am one of the authors of this review has some references on environmental sources of M. leprae and/or M. lepromatosis that I did not see included in your review. I am an not explicitly suggesting to include this article as a reference, its up to you if you find it useful or not. 
---

## [Decision Letter · Decision Letter 1]

7 Apr 2020

Dear Dr. Pieters,

We are pleased to inform you that your manuscript 'Reservoirs and Transmission Routes of Leprosy;

A Systematic Review' has been provisionally accepted for publication in PLOS Neglected Tropical Diseases.

Best regards,

Carlos Franco-Paredes

Associate Editor

Ana LTO Nascimento

Deputy Editor

Reviewer's Responses to Questions

**Key Review Criteria Required for Acceptance?**

**Methods**

-Are the objectives of the study clearly articulated with a clear testable hypothesis stated?

-Is the study design appropriate to address the stated objectives?

-Is the population clearly described and appropriate for the hypothesis being tested?

-Is the sample size sufficient to ensure adequate power to address the hypothesis being tested?

-Were correct statistical analysis used to support conclusions?

-Are there concerns about ethical or regulatory requirements being met?

Reviewer #1: (No Response)

Reviewer #2: The methods are well detailed and appropriate. They evidence a high degree of professional objectivity in the selection of papers for review.

Reviewer #3: (No Response)

**Results**

-Does the analysis presented match the analysis plan?

-Are the results clearly and completely presented?

-Are the figures (Tables, Images) of sufficient quality for clarity?

Reviewer #1: (No Response)

Reviewer #2: The paper is well researched and expertly presented.

Reviewer #3: Acceptable

**Conclusions**

-Are the conclusions supported by the data presented?

-Are the limitations of analysis clearly described?

-Do the authors discuss how these data can be helpful to advance our understanding of the topic under study?

-Is public health relevance addressed?

Reviewer #1: (No Response)

Reviewer #2: The authors have managed to draw clear logical conclusions about the status and importance of non-human reservoirs in leprosy with excellent objectivity.

Reviewer #3: (No Response)

**Editorial and Data Presentation Modifications?**

Reviewer #1: The synopsis has not been changed and needs to be edited to reflect the revisons made in the main manuscript.

Abstract - For improved English change the first sentence of paragraph 3 to read: More recently, M.leprae and M.lepromatosis DNA was deteced in red squirrels (Sciurus vulgaris) with leprosy-like lesions in teh British Isles.

Introduction 2nd paragraph : insert 'an' before environmental

Reviewer #2: None needed. Excellent paper.

Reviewer #3: (No Response)

**Summary and General Comments**

Reviewer #1: The authors have responded very satsfactorily to the reviewers comments resulting in a much -improved manuscript

Reviewer #2: This is a well written and constructed review by senior scholars with vast expertise in this area. It is likely to become a standard reference for both professionals and students. The topic is important to leprologists and others interested in neglected tropical diseases, and may help bring a better understanding about this disease for medical professionals and patients.

Reviewer #3: (No Response)

PLOS authors have the option to publish the peer review history of their article (what does this mean?). If published, this will include your full peer review and any attached files.

Reviewer #1: Yes: Anna Meredith

Reviewer #2: Yes: Richard Truman

Reviewer #3: No

---

## [Editor Report · Acceptance letter]

20 Apr 2020

Dear Prof. Dr. Pieters,

We are delighted to inform you that your manuscript, "Reservoirs and Transmission Routes of Leprosy;
A Systematic Review," has been formally accepted for publication in PLOS Neglected Tropical Diseases.

Best regards,

Serap Aksoy

Editor-in-Chief

Shaden Kamhawi

Editor-in-Chief
